# Patients with ANCA-Associated Glomerulonephritis and Connective Tissue Diseases: A Comparative Study from the Maine-Anjou AAV Registry

**DOI:** 10.3390/jcm8081218

**Published:** 2019-08-14

**Authors:** Fanny Guibert, Anne-Sophie Garnier, Samuel Wacrenier, Giorgina Piccoli, Assia Djema, Renaud Gansey, Julien Demiselle, Benoit Brilland, Maud Cousin, Virginie Besson, Agnès Duveau, Khuzama El Nasser, Jean-Philippe Coindre, Anne Croue, Jean-Paul Saint-André, Alain Chevailler, Jean-François Subra, Jean-François Augusto

**Affiliations:** 1CHU Angers, Service de Néphrologie-Dialyse-Transplantation, Université d’Angers, 49100 Angers, France; 2Service de Néphrologie, Centre Hospitalier du Mans, 72037 Le Mans, France; 3Service de Néphrologie, Centre Hospitalier de Cholet, 49300 Cholet, France; 4Service de Néphrologie, Centre Hospitalier de Laval, 53000 Laval, France; 5CRCINA, INSERM, Université de Nantes, Université d’Angers, 49100 Angers, France; 6CHU Angers, Département de Pathologie Cellulaire et Tissulaire, Université d’Angers, 49100 Angers, France; 7CHU Angers, Laboratoire d’Immunologie, Université d’Angers, 49100 Angers, France

**Keywords:** ANCA, vasculitis, connective tissue disease, glomerulonephritis

## Abstract

Background and objectives: The overlap between antineutrophil cytoplasmic antibody (ANCA) associated glomerulonephritis (ANCA-GN) and connective tissue diseases (CTD) has been reported mainly as case series in the literature. Frequency of this association, as well as presentation and outcomes are unknown. Materials and Methods: Patients from the Maine-Anjou ANCA-associated vasculitides (AAV) registry with ANCA-GN diagnosed between 01/01/2000 and 01/01/2018, ANCA positivity, and at least six months of follow-up, were included. Results: 106 out of 142 patients fulfilled the inclusion criteria and were analyzed. CTD was present at ANCA-GN diagnosis in 16 (15.1%) patients. The most common CTD were rheumatoid arthritis, Sjogren syndrome and systemic sclerosis. Compared to the control group, females were more represented in the CTD group (75%, *p* = 0.001). Renal presentation was comparable between groups, including the pathological analysis of renal biopsies. Patients of CTD group presented a higher rate of non-renal relapse (25% versus 7.7%, *p* = 0.037), and experienced more frequently a venous thrombotic event (31.2% versus 10%, *p* = 0.021). No difference between groups was observed according to major outcomes. Conclusion: Association between CTD and ANCA-GN is not a rare condition and predominantly affects females. While AAV presentation is not significantly different, CTD patients experience more frequently non-renal relapse and venous thrombotic events.

## 1. Introduction

Anti-neutrophil cytoplasmic antibody (ANCA)-associated vasculites (AAV) are autoimmune systemic diseases characterized by necrotizing inflammation of small to medium-sized vessels associated with the detection of myeloperoxidase (MPO) or proteinase-3 (PR3) -ANCAs in serum [1,2,3]. Three entities are differentiated on the basis of clinical and pathological criteria, with overlapping clinical spectra: microscopic polyangiitis (MPA), granulomatosis with polyangiitis (GPA), and eosinophilic granulomatosis with polyangiitis (EGPA) [4]. MPO-ANCAs are mainly observed in association with MPA, while PR3-ANCAs are more frequent in patients with GPA. AAV (GPA and MPA) are rare diseases, with estimated incidence between 13 and 20/million per year and an estimated prevalence between 46 and 184/million depending on ethnicity and geographical factors [5]. Gender distribution is equilibrated, and the peak incidence age is increasing and between 65 and 75 years old [5].

AAV are life-threatening diseases responsible for the burden of high morbidity and mortality rates related to vasculitis, but also to the consequences of the long-term immunosuppressive treatment [6]. Renal involvement is a major prognostic factor for death and indicates the need to start an aggressive immunosuppressive treatment immediately [6]. Pauci-immune crescentic necrotizing glomerulonephritis is the most frequent histological feature, common to GPA and MPA, and accounts for the rapidly and progressive deterioration of renal function. Presently the most widely used induction remission treatment of ANCA-glomerulonephritis (ANCA-GN) combines steroids with cyclophosphamide or rituximab [6]. By these treatments, complete remission is achieved in most patients within three to six months, allowing patients to start a maintenance regimen that aims at preventing AAV relapse.

The association of AAV with other auto-immune diseases or connective tissue diseases (CTD) such as systemic lupus erythematosus (SLE), rheumatoid arthritis (RA) or Sjogren syndrome (SS) is a rare, but not unheard-of condition. Indeed, such associations have been reported in the literature in forms of case reports or case series [7,8,9,10,11]. However, published data do not allow the prevalence of this overlapping syndrome and the nature of CTD subtypes observed in these patients to be concluded. Moreover, the outcome of this subgroup of patients in comparison to other AAV patients has not yet been analyzed.

The aim of the present study was to analyze the frequency and outcome of patients with AAV previously diagnosed with CTD. For this, we used the AAV Maine-Anjou Registry that includes all consecutive AAV patients from the Nephrology Departments of one University Hospital and three Regional Hospitals. We focused on patients that presented with new onset or relapsing AAV with ANCA-GN at diagnosis. The CTD + AAV group was compared to a control AAV group without CTD with regard to AAV presentation and outcome.

## 2. Materials and Methods

### 2.1. Study Population

All consecutive AAV patients with pauci-immune glomerulonephritis diagnosed between 01/01/2000 and 01/01/2018 were identified from the Maine-Anjou AAV registry (registry of patients with AAV admitted to the Nephrology Departments of Angers University hospital and of the Regional Hospitals of Le Mans, Cholet and Laval) and were included in the present study. Patients were eligible if they were aged > 18 years-old, fulfilled the Chapel Hill Consensus Conference criteria for AAV [4], presented with pauci-immune glomerulonephritis at kidney biopsy, positives MPO or PR3 ANCAs and had at least 6 months of follow-ups. ANCA negative patients were excluded. Only the histological data of the first kidney biopsy was considered in the present study.

The study protocol agreed with the Ethics Committee of the Angers University Hospital (authorization number, 2019/06; authorization date, 02/19/2019). All participants of the study gave their oral consent to participate in the registry. Patient records and related data were anonymized prior to analysis.

### 2.2. Data Collection

Data were collected retrospectively after individual screening of the patients’ medical records. The following data were retrieved: age, gender, weight, and significant medical history. The nature and type of injuries to the affected organs upon AAV presentation were listed. The activity of AAV and organ involvement at diagnosis and relapse were determined in reference to the Birmingham Vasculitis Activity Score (BVAS) 2003 [12]. Glomerular filtration rate was calculated using the 4-variable Modification of Diet in Renal Disease (MDRD) study equation [13]. Therapeutic regimens, retrieved from medical records, were comparable between centers, although they evolved according to AAV standard of care.

### 2.3. Definitions

Patient records were analyzed and AAV subtype (GPA and MPA) was determined according to the European Medicines Agency (EMA) vasculitis classification algorithm [14]. The identification of renal disease was based on clinical data (active urinary sediment, proteinuria and impaired renal function) and kidney biopsies. Renal death was defined as the need for renal replacement therapy (RRT) for more than 3 months. Relapse was defined as the recurrence or new signs of organ involvement attributable to AAV activity, requiring the need to increase steroids or to start an immunosuppressive treatment. We individualized renal and non-renal relapses. Renal relapses were defined as relapses involving the kidneys, characterized by new or worsening red blood casts and/or worsening proteinuria in association with a rise in serum creatinine of at least 25%. Non-renal relapses were defined as AAV activity in any other organ, solely or in combination, but excluding kidneys.

The diagnosis of CTD was verified according to relevant classifications. In brief, the ACR/European League Against Rheumatism (EULAR) [15], the ACR [16,17], the 2016 American-European consensus group [18], and the ACR/EULAR 2013 [19] classifications were used for the diagnosis of RA, SLE, GSS and systemic sclerosis (SSc), respectively.

Severe infectious events were defined as infections needing hospital admission. The following cardio-vascular events were retrieved from patients’ records: episodes of myocardial infarction and coronary revascularization, brain strokes, and surgical or endovascular aortic or peripheral artery revascularization. Venous thrombotic events confirmed by CT-scan or Doppler ultrasound were also retrieved.

### 2.4. Renal Histopathology

All kidney biopsies from the four centers were analyzed centrally in the department of Pathology of the University Hospital of Angers by JPSA and AC. Patients whose biopsy showed less than 10 glomeruli in a light microscopy analysis were excluded from the study. The microscopic analysis of kidney biopsies is routinely reported in a standardized pathological report. Kidney biopsies were classified in the histopathological classification of ANCA-GN [20] and immune deposits by immunofluorescence analysis were quantified as follows: 0, none; 1+, weak; 2+, moderate; 3+, strong.

### 2.5. Statistical Analysis

Continuous variables are presented as means ± SD and categorical variables as the absolute value and percentage. Differences between groups were analyzed using the χ^2^ test (or Fisher exact test where applicable) for categorical variables and the Mann-Whitney U test for continuous variables. All the statistical tests were performed to the two-sided 0.05 level of significance. Statistical analysis was performed using SPSS software® 23.0 for Mackintosh (IBM, Armonk, NY, USA) and Graphpad Prism^®^ (San Diego, CA, USA).

## 3. Results

### 3.1. Baseline Characteristics of Patients of the CTD Group

Among the patients on the registry, 106 fulfilled the inclusion criteria for the study (Figure 1). A diagnosis of CTD was observed at AAV diagnosis in 16 (15.1%) out of 106 successive AAV patients. Patients with CTD were female in 12/16 (75%) cases and their mean age at AAV diagnosis was 65 years old. The main CTD diagnosis was rheumatoid arthritis (n = 5), systemic sclerosis associated and/or Sjogren syndrome (n = 4) and polymyalgia rheumatic (n = 3). The CTD diagnosis preceded the onset of AAV by 7 ± 6.1 years. Ten patients (62.5%) were on steroids or immunosuppressive treatment by the time they were diagnosed with AAV. These data are summarized in Table 1. Figure 2 shows the incidence of AVV and of CTD combined with AAV.

### 3.2. Baseline Characteristics of AAV Patients According to the Presence or Absence of CTD

Compared to the control group (AAV patients), females were more frequently represented in the CTD group (*p* = 0.001). The mean age between groups was comparable. There was also no difference according to AAV phenotype or ANCA subtype distribution. Organ involvement at AAV onset did not differ between groups and mean BVAS was comparable. Induction remission treatment relied predominantly on cyclophosphamide and 30% of patients from both groups received plasma exchange. These data are detailed in Table 2.

Anti-nuclear antibodies were detected in 80% of the patients of the CTD group, compared to 47.5% in the control group (*p* = 0.021, Table 3). A type 3 cryoglobulin positivity was observed in 3 (37.5%) patients in the CTD group, compared to 2 (5.7%) patients in the control group (*p* = 0.011).

### 3.3. Comparison of Renal Presentation According to Groups

Although, according to renal involvement at the time of kidney biopsy, the CTD group tended to have lower levels of serum creatinine, eGFR was comparable between groups (Table 2). None of the patients of the CTD group required renal replacement therapy (RRT) at kidney biopsy, while RRT was needed in 15.5% of patients in the AAV group. However, this was not statistically significant (*p* = 0.21).

The analysis of renal biopsy did not show any significant difference between groups. The percentages of normal, crescentic and fibrotic glomeruli were comparable (Figure 3A), although the patients of the CTD group tended to have a lower percentage of fibrotic glomeruli. When classified according to the histopathological classification of AAV, the distribution of classes was also comparable between groups (Figure 3B).

Glomerular immune deposits in immunofluorescence studies of renal biopsies did not reveal any difference between groups (Table 4).

### 3.4. Comparison of Outcomes According to Groups

Next, we analyzed the rate and nature of AAV relapses in each group (Table 5). First, the maintenance regimens were compared between groups, with two thirds of patients receiving azathioprine and one third receiving rituximab. The CTD group tended to experience relapses more frequently than the control group (*p* = 0.096). The rate of non-renal relapse was significantly higher in the CTD group compared to the control group (*p* = 0.037), while the rate of renal relapses did not differ. The median delay to relapse was 29.5 months in the CTD group and 46.1 months in the control group. The mean dose of steroids between month 6 and month 24, as well as the rate of patients with steroid withdrawal, did not differ between groups.

In a last step, we analyzed other major events including deaths, severe infections requiring hospital admission, cardiovascular events, cancers and venous thrombotic complications (Table 6). We did not observe any difference between the CTD group and the control group with regard to death, cancer, infectious events and cardiovascular events. However, thrombotic events occurred more frequently in the CTD group, affecting 31.2% of patients during their follow-up, compared to 10% in the control group (*p* = 0.021). When major events were grouped within a composite criterion including death, infections, cardiovascular events cancers and thrombotic events, no difference between groups was observed.

## 4. Discussion

In the present work, we show that approximately 15% of consecutive AAV patients have a diagnosis of CTD, showing that overlap syndrome is in fact not an uncommon condition in AAV patients. Our study failed to identify any difference in respect to AAV presentation, including renal presentation, between CTD patients and control patients. The main finding of this study is that, despite a higher rate of non-renal AAV relapse and venous thrombosis, CTD patients seem to have a comparable prognosis according to a composite criterion, including major events (deaths, severe infections, cancer, cardiovascular events and thrombotic events). To the best of our knowledge, this study is the first to enable the relative frequency of CTD among AAV patients to be estimated and to analyze its impact on them in comparison to a control group.

The most common CTD observed in the AAV patients of the present cohort study were RA, SS and SSc. Interestingly, two patients had an overlapping syndrome associating SSc and SS. Associations between CTD and AAV have mainly been reported as case reports and series in the literature, not allowing the frequency of such associations to be determined. In a recently published study, the frequency of the association between systemic auto-immune diseases and AAV was 11.3%, which is very close to our observation (15.1%) [10]. Interestingly, as in our work, RA, SSc and SS were the most frequently observed CTD. Only one patient was diagnosed with SLE and AAV in our study. While ANCA detection is quite frequent in SLE patients, the association between SLE and AAV seems to be rare. In support, a nationwide French study, only eight patients with both diseases were identified [9]. Recent findings of associations between gene polymorphisms and the risk of AAV may explain why some patients are prone to developing two auto-immune diseases. Indeed, some polymorphisms have been linked to both AAV and CTD development [21,22]. As an example, polymorphisms of the PTPN22 gene have been shown to favor both GPA and RA occurrence [22]. Another explanation may be the exposure to risk factors common to several auto-immune diseases, such as environmental factors or exposure to certain drugs. The pathophysiological connection is also well illustrated by the frequently lowered immune tolerance against ANCA antigens observed in CTD patients. Indeed, ANCA positivity, not only against MPO or PR3-ANCAs, but also against minor ANCA antigens (i.e., lactoferrin, BPI) is frequently observed in CTD. ANCA positivity, with mainly atypical ANCA patterns, has been observed in up to 15% in RA [23,24], 15–30% in SLE [25,26], 35% in SSc [27], and 10% in SS patients [28,29]. However, only a minority of ANCA positive CTD patients will finally develop AAV. These observations suggest the involvement of common pathophysiological auto-immune pathways in the development of AAV and CTD. As expected, significantly more patients were diagnosed with cryoglobulin positivity in CTD group, however, no difference was observed according to rheumatoid factor positivity between groups. This may be explained by a high rate of false positivity of rheumatoid factor detected in AAV patients [30].

We observed that overlap between AAV and CTD occurred more frequently in females, compared to the control group, which was also observed in a previous study [10]. This may result from the higher predisposition of women to developing most of the CTD observed in the present cohort, rather than a specific factor predisposing women to the development of AAV [31].

The present study is the first to compare presentation and outcomes for patients with AAV+CTD to a control group. Apart from sex ratio, we did not observe any other baseline difference between the CTD and control groups. AAV phenotype and ANCA specificity did not differ significantly. This observation does not corroborate the findings from previous case reports and studies, which suggested a predominance of MPA phenotype and MPO ANCA positivity in CTD patients [7,10,28]. We did not observe any difference in AAV organ involvement nor in disease activity between groups at AAV onset. Concerning renal involvement, no difference was observed with regards to kidney function and histological involvement between groups. This suggests that CTD per se may not be a factor able to modulate glomerular auto-immune injury in AAV.

The present work suggests that there are different outcomes for AAV patients with CTD. Indeed, we observed a higher rate of extra-renal AAV relapse in CTD patients compared to the control group, but no difference in the renal and overall relapse rate was observed. This was not likely due to differences in treatment, as both remission induction and maintenance regimens were comparable between groups, including steroid treatment management. The small size of the population and the low number of relapses did not allow to perform multivariate analysis and to adjust on gender and renal function. However, this observation suggests that maintenance regimens in patients with CTD should be carefully monitored and their intensity and duration eventually adjusted. As non-renal relapses involved mainly minor relapses, we also acknowledge that we could have misdiagnosed them as AAV relapses although they could reflect CTD activity.

A major finding of the present work is that the rate of major events was comparable between groups. The only difference was a higher rate of venous thrombotic events in the CTD group, which occurred in more than one third of patients during the follow-up. Discussing the mechanisms accounting for this observation is beyond what we can infer from the present study, but we can suggest that this could be related to a more inflammatory state related to CTD. Thus, AAV+CTD patients should be carefully monitored for thrombosis and prophylactic treatment should be considered more easily in at risk situations for these patients. Finally, after analyzing single and pooled major events, we did not observe any difference between CTD and CTD+AAV patients, suggesting that prognosis does not differ.

The limitations of our study lie in its retrospective design. The follow-up of patients, about 4.5 years, may also represent a limitation of the study. Moreover, we acknowledge that the CTD group is heterogeneous in nature given the multiplicity of CTD and pooling them within one group may be controversial. However, we think that our study adds new knowledge to the field of AAV. By analyzing successive and histologically confirmed AAV, we were able to accurately estimate the prevalence of CTD in AAV patients. Moreover, the Maine-Anjou AAV registry includes data from four regional nephrology centers with very similar treatments and follow-ups between centers.

## 5. Conclusions

In conclusion, CTD in association with ANCA-GN diagnosis is not a rare condition, predominantly affecting women and without any significant differences in AAV presentation. This study suggests that, despite a comparable prognosis, CTD patients have a higher risk of non-renal AAV relapse and of venous thrombotic events.

## Figures and Tables

**Figure 1 jcm-08-01218-f001:**
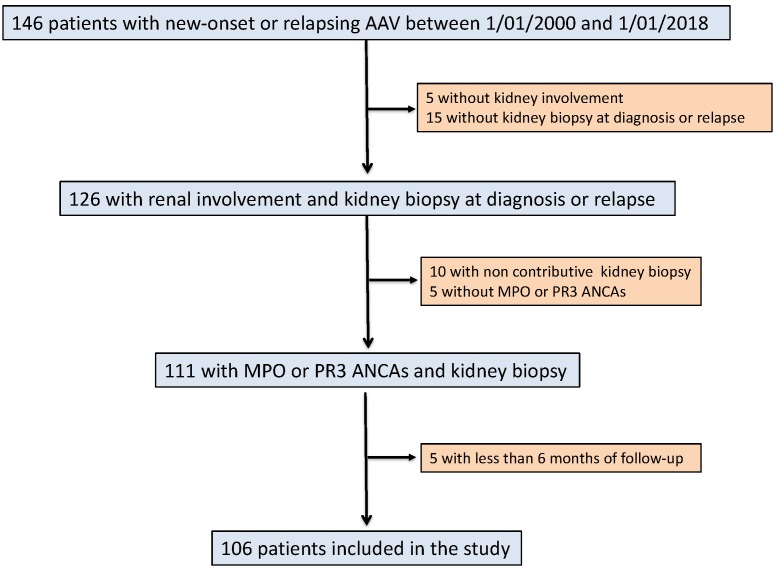
Flowchart of the study. AAV, ANCA-associated vasculitis; ANCA, antineutrophil cytoplasmic antibody; MPO, myeloperoxidase; PR3, proteinase-3.

**Figure 2 jcm-08-01218-f002:**
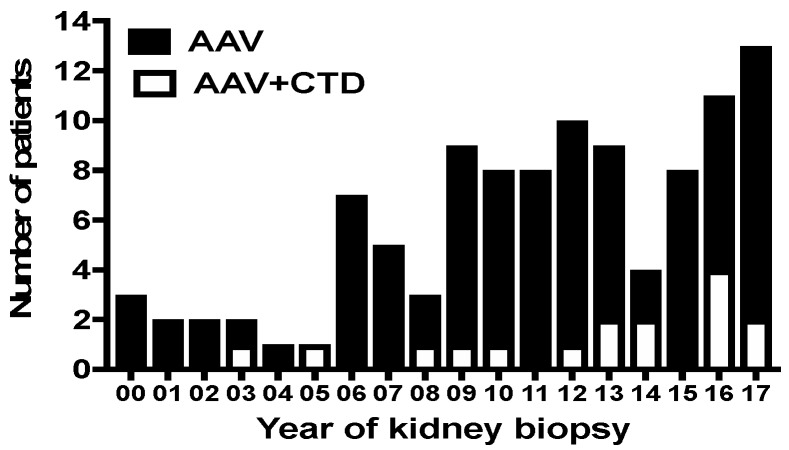
Incidence of ANCA-associated vasculitides (AAV) and AAV+CTD in the cohort study according to the year of diagnosis. AAV, ANCA-associated vasculitis; CTD, connective tissue disease.

**Figure 3 jcm-08-01218-f003:**
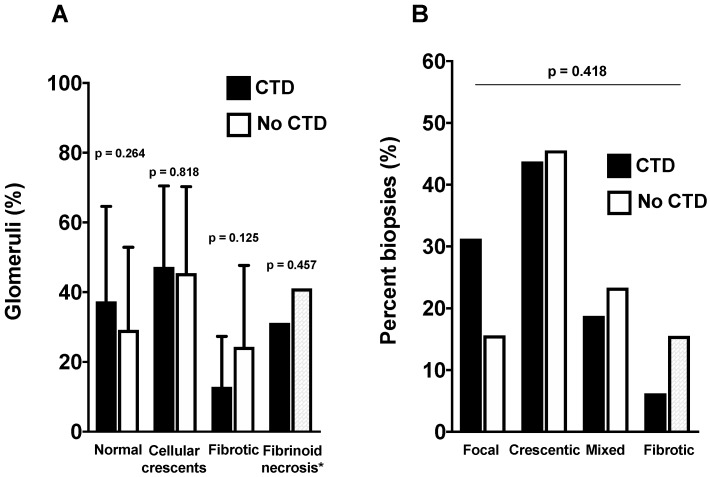
Analysis of light microscopy of kidney biopsy and comparison according to groups. (**A**) percentage of normal, crescentic and fibrotic glomeruli according to the presence or absence of CTD. (**B**) histopathological classes of ANCA-GN according to groups.

**Table 1 jcm-08-01218-t001:** Main characteristics of patients with connective tissue diseases (CTD).

Patients (P)	Gender	CTD Diagnosis	Age at CTD	Delay *	CTD-Related Organ Involvement	IS **	Nature of IS	IS at AAV Onset
P1	Female	RA	54	10 y	Joints	Yes	MTX	No
P2	Female	RA	72	2 y	Joints	Yes	Steroids	Yes (steroids)
P3	Male	RA	41	8 y	Joints	Yes	Steroids, MTX, anti-TNF	Yes (anti-TNF)
P4	Female	RA	65	6 y	Joints	Yes	Steroids, MTX, anti-TNF	Yes (anti-TNF)
P5	Female	RA	35	10 y	Joints	Yes	Steroids, MTX, anti-TNF	Yes (anti-TNF)
P6	Female	Sarcoidosis	57	9 y	Lung, nodes, salivary glands	Yes	Steroids	No
P7	Male	SSc	71	1 y	Skin, Sicca syndrome	No	/	No
P8	Female	SSc + SS	61	3 y	Skin, lung, Sicca syndrome	Yes	Steroids	Yes (steroids)
P9	Female	SSc + SS	49	6 y	Skin, joints, Raynaud, Sicca syndrome	No	/	No
P10	Male	SS	52	25 y	Lung	Yes	Steroids	No
P11	Female	SLE	44	5 y	Joints	Yes	Steroids, HCQ	Yes (steroids)
P12	Female	PR	74	3 y	Joints	Yes	Steroids, MTX	Yes (steroids+MTX)
P13	Female	PR	60	2 y	Joints	Yes	Steroids	Yes (steroids)
P14	Female	PR	74	1 y	Joints	Yes	Steroids	Yes (steroids)
P15	Male	RP	60	7 y	Joints, ear, nose	Yes	Steroids, MTX	Yes (steroids+MTX)
P16	Female	PA	50	14 y	Joints, skin	Yes	Methotrexate	No

RA, Rheumatoid arthritis; SSc, systemic sclerosis; SS, Sjogren syndrome; PR, Polymyalgia rheumatica; RP, Relapsing polychondritis; PA, Psoriatic arthritis; IS, immunosuppressive treatment; MTX, methotrexate; HCQ, Hydroxychloroquine. * between CTD and AAV diagnosis in years; ** before AAV diagnosis

**Table 2 jcm-08-01218-t002:** Baseline characteristics of AAV patients according to presence or absence of CTD. Significant *p* values appear in bold.

Variables	All, n = 106	Connective Tissue Diseases		
		Yes, n = 16	No, n = 90	*p*
**Baseline characteristics**				
Sex (M/F)	67/39	4/12	63/27	**0.001**
Age (years)	63.4 ± 14.0	65 ± 10.3	63.1 ± 14.6	0.619
**ANCA-associated vasculitis characteristics**				
**Clinical diagnosis**, n (%)				
GPA/MPA	36 (34)/70 (66.6)	7 (43.7)/9 (56.3)	29 (32.2)/61 (67.8)	0.37
Newly diagnosed	98 (92.4)	15 (93.7)	83 (92.2)	0.831
**ANCA type**, n (%)				
c-ANCA/p-ANCA, n (%)	32 (30.2)/74 (69.8)	7 (43.7)/9 (56.3)	25 (27.8)/65 (72.2)	0.2
PR3-ANCA/MPO-ANCA, n (%)	32 (30.2)/74 (69.8)	7 (43.7)/9 (56.3)	25 (27.8)/65 (72.2)	0.2
**BVAS at kidney biopsy**	17.3 ± 5.7	15.6 ± 5.3	17.6 ± 5.8	0.2
**Organ involvement at diagnosis**, n (%)				
Cutaneous signs	20 (18.9)	3 (18.7)	17 (18.9)	1
Ear, nose, throat	34 (32.1)	6 (37.5)	28 (31.1)	0.772
Heart	6 (5.7)	1 (6.2)	5 (5.6)	1
Digestive	4 (3.7)	1 (6.2)	3 (3.3)	0.486
Lung	40 (37.7)	4 (25)	36 (40)	0.401
Renal (at kidney biopsy)	-			
Serum creatinine, µmol/L	350.2 ± 296	264.7 ± 188	364.9 ± 309	0.227
eGFR, mL/min/1.73 m^2^	32.5 ± 34.7	35.0 ± 30.2	32.1 ± 35.5	0.766
Need for renal replacement therapy, n (%)	13 (12.3)	0 (0.0)	13 (15.5)	0.21
Neurological	14 (13.2)	2 (12.5)	12 (13.3)	0.928
**Induction therapy**, n (%)				
Cyclophosphamide	97 (91.5)	14 (87.5)	83 (92.2)	0.663
Rituximab	3 (2.8)	1 (6.2)	2 (2.2)	-
Other	6 (5.7)	1 (6.2)	5 (5.5)	-
Plasma exchange	31 (29.2)	5 (31.2)	26 (28.8)	0.848

**Table 3 jcm-08-01218-t003:** Immunological presentation according to groups. Significant *p* values appear in bold.

	Connective Tissue Disease		
	Yes, n = 16	No, n = 90	*p*
**Antinuclear antibody**			
Screened patients, n (%)	15 (93.7)	80 (88.8)	0.556
Positives (≥ 1/100), n (%)	12 (80)	38 (47.5)	**0.021**
≥ 1/200, n (%)	8 (50)	25 (31.2)	**0.036**
**Antigen specificity, number of patients**	4	2	/
Anti-SSA, n	1	0	/
Anti-SSB, n	0	0	/
Anti-centromere, n	2	1	/
Anti-mitochondrial, M2 subtype, n	1	1	/
**Anti-GBM antibody**			
Screened patients, n (%)	15 (93.7)	71 (78.8)	0.161
Positives patients, n (%)	1 (6.3)	1 (1.1)	0.32
**Cryoglobulin**			
Screened patients, n (%)	8 (50)	35 (38.9)	0.402
Positive, n (%)	3 (37.5)	2 (5.7)	**0.011**
**Rheumatoid factor**			
Screened patients, n (%)	5 (31.2)	18 (20)	0.314
Positive, n (%)	4 (80)	7 (38.9)	0.103

**Table 4 jcm-08-01218-t004:** Immunofluorescence analysis of kidney biopsy according to study groups. Deposits were graded as follows: 0, none; 1+, weak; 2+, moderate; 3+, strong.

Immunofluorescence Study	All, n = 106	Connective Tissue Disease		
		Yes, n = 16	No, n = 90	*p*
IgG deposits	0.12 ± 0.4	0 ± 0	0.14 ± 0.5	0.24
≥2+, n (%)	2 (1.9)	0 (0)	2 (2.2)	1
IgA deposits	0.14 ± 0.5	0 ± 0	0.17 ± 0.5	0.229
≥2+, n (%)	2 (1.9)	0 (0)	2 (2.2)	1
IgM deposits	0.89 ± 0.5	1 ± 0.5	0.86 ± 0.5	0.343
≥2+, n (%)	7 (6.6)	2 (12.5)	5 (5.5)	0.284
C1q deposits	0.2 ± 0.5	0.2 ± 0.4	0.17 ± 0.5	0.821
≥2+, n (%)	1 (0.9)	0 (0)	1 (1.1)	1
C3 deposits	0.63 ± 0.8	0.4 ± 0.7	0.67 ± 0.7	0.202
≥2+, n (%)	11 (10.4)	2 (12.5)	9 (10)	0.67

**Table 5 jcm-08-01218-t005:** Characteristics of relapses and maintenance regimen according to groups. Significant p-values appear in bold.

Variables	All, n = 106	Connective Tissue Disease		
		Yes, n = 16	No, n = 90	*p*
**Relapses, all, n (%)**	23 (21.7)	6 (37.5)	17 (18.8)	0.096
Mean delay (months)	43.6 ± 43.2	29.5 ± 30.1	46.1 ± 44.8	0.158
**Non-renal relapse, n (%)**	11 (10.4)	4 (25)	7 (7.7)	**0.037**
Mean delay (months)	35.4 ± 39	24.6 ± 34	41.6 ± 42.9	0.518
**Renal relapse, n (%)**	12 (11.3)	2 (12.5)	10 (11.1)	1
Mean delay (months)	49.1 ± 37.7	35.7 ± 12.3	51.7 ± 40.9	0.607
**Steroids**				
Steroids at Month 6	12.5 ± 8.9	12.3 ± 4.9	12.5 ± 9.6	0.935
Steroids at Year 1	5.5 ± 5.4	4.5 ± 4	5.7 ± 5.6	0.486
Steroids at Year 2	3.4 ± 9.5	1.93 ± 2.9	3.7 ± 10.1	0.638
Steroid withdrawal	72 (67.9)	12 (75.0)	60 (66.6)	0.753
Mean delay	18.4 ± 19.1	15.7 ± 15.3	19.0 ± 19.9	0.604
**Maintenance regimen, n (%)**				
Azathioprine	68 (64.2)	10 (62.5)	58 (64.4)	0.881
Rituximab	33 (31.1)	6 (37.5)	27 (30)	0.55
Other	5 (4.7)	0 (0)	5 (5.6)	-

**Table 6 jcm-08-01218-t006:** Outcomes according to study groups.

Events	All, n = 106	Connective Tissue Disease		
		Yes, n = 16	No, n = 90	*p*
**Death, n (%)**	16 (15.1)	2 (12.5)	14 (15.5)	1
Mean delay (months)	32 ± 28.2	37.8 ± 13.2	31.2 ± 29.8	0.767
**Renal function**				
eGFR at year 1, mL/min/1.73 m^2^	42.1 ± 28.4	38.8 ± 27.3	42.6 ± 28.7	0.709
eGFR at year 3, mL/min/1.73 m^2^	44.2 ± 32.6	55.1 ± 14.6	42.9 ± 34.1	0.435
End-stage renal disease, n (%)	33 (31.1)	5 (31.3)	28 (31.1)	0.991
Mean delay (months)	19.5 ± 27.4	22.6 ± 20.6	18.9 ± 28.7	0.789
**Severe infectious events, n (%)**	48 (45.3)	7 (43.7)	41 (45.5)	0.894
Mean delay (months)	23.8 ± 39.1	13.7 ± 16	25.7 ± 41.9	0.461
**Cardiovascular events, n (%)**	12 (11.3)	1 (6.25)	11 (12.2)	0.688
Mean delay (months)	57.1 ± 56.1	-	62.3 ± 55.9	-
Myocardial infarction, n (%)	1 (0.9)	1 (6.25)	0 (0)	0.151
Stroke, n (%)	2 (1.9)	0 (0)	2 (2.2)	1
Others, n (%)	9 (8.5)	0 (0)	9 (10)	0.349
**Cancer, n (%)**	13 (12.3)	1 (6.2)	12 (13.3)	0.686
Mean delay (months)	40.9 ± 40	-	40.9 ± 40	-
Solid cancer, n (%)	6 (5.7)	0 (0)	6 (6.7)	0.588
Skin cancer, n (%)	7 (6.6)	1 (6.2)	6 (6.7)	0.934
**Thrombotic events, n (%)**	14 (14.2)	5 (31.2)	9 (10)	**0.021**
Mean delay (months)	47.2 ± 11.9	17.1 ± 17.7	9.03 ± 16.4	0.461
**At least one event, n (%) ***	66 (62.3)	11 (68.7)	55 (61.1)	0.302
Mean follow-up **(months)**	55.6 ± 52.2	40.6 ± 40.6	58.3 ± 53.8	0.213

* Death, severe infection, cardiovascular event, cancer or thrombotic event. Significant *p* values appear in bold.

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
