# Peer review of "Patients with ANCA-Associated Glomerulonephritis and Connective Tissue Diseases: A Comparative Study from the Maine-Anjou AAV Registry"

_jcm, 2019, doi:10.3390/jcm8081218_

Round 1
Reviewer 1 Report
Dear editors:
Jean-François Augusto and co-authors conducted a clinical research of “Patients with ANCA-associated Glomerulonephritis and connective tissue diseases: a comparative study from the Maine-Anjou AAV registry”. I have a number of questions and comments concerning this study:
1. Considering the description of “Delay*” in Table 1, the explanation of “* between CTD and AAV diagnosis, in years, ** before AAV diagnosis” is confusing. The symbol ** can’t be found in Table 1. According to Figure 1 (Flowchart of the study) and Table 2, 16 cases of CTD may be diagnosed based on chart review after the renal biopsy. The sentence of “* between CTD and AAV diagnosis, in years, ** before AAV diagnosis” should be rephrased. As shown in Figure 1, all study cases of AAV with biopsy -proven renal involvement should be expressed as ANCA-associated Glomerulonephritis (ANCA-GN).
2. Compared with previous research, there are conflicting results. First, the prevalence of C-ANCA in Table 3 showed 7/16 (43.7%), which is an unusual result. The prevalence of ANCA antibodies in a more larger inception cohort study showed only P-ANCA without C-ANCA detection in CTD patients after final interpretation (Ann Intern Med. 1997 Jun 1;126(11):866-73.). And high percentage patients with systemic lupus erythematosus (SLE) have ANCA-GN + CTD with positive P-ANCA (Ann Intern Med. 1997 Jun 1;126(11):866-73.) (Case Reports in Rheumatology, vol. 2016, Article ID 9070487, 6 pages, 2016.). After literature review, although the dual presentation of ANCA-GN and CTD are rare case reports, all of such cases are P-ANCA (Ann Rheum Dis. 2006 Mar; 65(3): 410–411) (Intern Med. 2018 Jun 15; 57(12): 1757–1762.) (Rheumatol Int. 2016 Sep;36(9):1327-34.)( J Formos Med Assoc. 2011 Jul;110(7):473-7.)(….). Could you explain why the extremely high prevalence of C-ANCA exists in your study? Also, it is unusual that SLE could not be found in the 16 cases of ANCA-GN + CTD.
3. Second, compared with the final result interpretation of 2.8% positive Anti-MPO (1.4% SLE and 1.4% RA) in previous study (Ann Intern Med. 1997 Jun 1;126(11):866-73.), it is unusual that such a high prevalence 15.1% patients have ANCA-GN + MTD.
4. Third, ANCA GN with CTD often correlates with disease severity and portends a poorer prognosis (Mittie K. Doyle. Current Rheumatology Reports 2006, 8:312–316), whilst Table 6 and the conclusion in this study suggest a comparable long term prognosis between both groups. The analysis of renal biopsy did not show any significant difference between groups (Figure 3), which is inconsistent with previous studies. Considering the low event rate in a very small sample size study, the statistical analysis is a critical issue.
5. In Table 2, the need for renal replacement therapy [n (%)]: 0 (0.0) in AAV + MTD group VS. 13 (15.5) in AAV group, the p value is non-significant (p=0.21)?
I think AAV group is significantly associated with the need for renal replacement therapy. With respect to these cases with initial renal replacement therapy, renal relapses should not be comparable between groups.
6. What are the definitions of renal and non-renal relapses? All patients with ANCA GN should receive strong immunosuppressive therapy and plasmapheresis/ plasma exchange, etc. Such standard treatments were specific to not only ANCA-GN but also CTD, as these agents were powerful and effective to prevent various relapses. Since renal relapses were comparable, please address why non-renal relapses were more frequent in ANCA GN + CTD group.
7. In light of Figure 2, the white bar was expressed as incidence of AAV + CTD case number according to the year of diagnosis (biopsy-proven), and the black bar as AAV. However, 6 cases of AAV+CTD were diagnosed in 2016-2017. In these 6 cases, I wonder whether the follow up period is long enough to observe the events of deaths, cardiovascular outcomes, cancers and so on. Considering the low event rate in a very small sample size study, the statistical power is another issue.
8. The consensus of current research indicates that dual presentation of ANCA-GN and CTD is rare, including authors' description in the Introduction section. However, the authors concluded that CTD is “frequently” associated with ANCA-GN. In my opinion, the term “frequently” should be deleted.
9. Drug-induced ANCA-GN is an important issue. Is there any difference between groups about secondary causes related ANCA-GN?
Author Response
Response to Reviewer 1 comments.
We acknowledge Reviewer 1 for his valuable comments.
Dear editors:
Jean-François Augusto and co-authors conducted a clinical research of “Patients with ANCA-associated Glomerulonephritis and connective tissue diseases: a comparative study from the Maine-Anjou AAV registry”. I have a number of questions and comments concerning this study:
Considering the description of “Delay*”in Table 1, the explanation of “* between CTD and AAV diagnosis, in years, ** before AAV diagnosis” is confusing. The symbol ** can’t be found in Table 1. According to Figure 1 (Flowchart of the study) and Table 2, 16 cases of CTD may be diagnosed based on chart review after the renal biopsy. The sentence of “* between CTD and AAV diagnosis, in years, ** before AAV diagnosis” should be rephrased. As shown in Figure 1, all study cases of AAV with biopsy -proven renal involvement should be expressed as ANCA-associated Glomerulonephritis (ANCA-GN).
We totaly agree with reviewer 1. This is a typing error, the ** refers to IS treatment. This was corrected within the Table 1.
Compared with previous research, there are conflicting results. First, the prevalence of C-ANCA in Table 3 showed 7/16 (43.7%), which is an unusual result. The prevalence of ANCA antibodies in a more larger inception cohort study showed only P-ANCA without C-ANCA detection in CTD patients after final interpretation (Ann Intern Med. 1997 Jun 1;126(11):866-73.). And high percentage patients with systemic lupus erythematosus (SLE) have ANCA-GN + CTD with positive P-ANCA (Ann Intern Med. 1997 Jun 1;126(11):866-73.) (Case Reports in Rheumatology, vol. 2016, Article ID 9070487, 6 pages, 2016.). After literature review, although the dual presentation of ANCA-GN and CTD are rare case reports, all of such cases are P-ANCA (Ann Rheum Dis. 2006 Mar; 65(3): 410–411) (Intern Med. 2018 Jun 15; 57(12): 1757–1762.) (Rheumatol Int. 2016 Sep;36(9):1327-34.)(J Formos Med Assoc. 2011 Jul;110(7):473-7.)(….). Could you explain why the extremely high prevalence of C-ANCA exists in your study? Also, it is unusual that SLE could not be found in the 16 cases of ANCA-GN + CTD.
We acknowledge Reviewer 1 for these important remarks. We completely agree with reviewer 1 that our findings are not in line with what has been published previously. However, we think that the results of our study cannot be compared with the observations of previous papers, for many reasons, detailed below:
In the present work, we analyzed the association between CDT and AAV-GN. Importantly, AAV diagnosis was based on clinical and histological data. We didn’t analyze the association between CTD and ANCA serology, as done in the study of Merkel P et al (Ann Intern Med. 1997 Jun 1;126(11):866-73) . As reviewer 1 knows and underlined, the association between auto-antibodies and CDT, do not allow to conclude to any pathogenicity of ANCAs in these patients with CTD. One of the strengths of our study is that we analyzed in a non-biased manner all consecutive patients with histologically-proved ANCA-GN and looked for those patients with CDT in a multicenter study, covering a population of 3 million peoples. Thus we believe that this constitutes the appropriate methodology to analyze prevalences. We agree that based on case reports or case series, patients reported in the literature with AAV and CTD were predominantly pANCA and MPO ANCA positives. However, even if we didn’t observe a statistical difference between C and P ANCAs in our study, a predominance of p-ANCA of MPO specificity were also observed. In respect to SLE, even if ANCA detection is very prevalent, the association of SLE and AAV is a quite rare clinical condition. Tolerance break to a multitude of self-antigens is a hallmark of SLE and we agree that a clear predominance of MPO ANCAs are detected in SLE. However, the association of SLE and AAV is a very rare condition. In support, a previous and recent French nationwide study identified only 8 patients with this condition (Jarrot PA et al, Medicine, volume 95, number 22, 2016). Thus, it is finally not so surprising that only one patient with SLE and AAV was identified in our study. Interestingly, this patient was MPO ANCA positive.
We added the following precision to the discussion to underline these points (page 8, line 214):
“Only one patient was diagnosed with SLE and AAV in our study. While ANCA detection is quite frequent in SLE patients, the association between SLE and AAV seems to be rare. In support, a nationwide French study, only 8 patients with both diseases were identified [9].”
Second, compared with the final result interpretation of 2.8% positive Anti-MPO (1.4% SLE and 1.4% RA) in previous study (Ann Intern Med. 1997 Jun 1;126(11):866-73.), it is unusual that such a high prevalence 15.1% patients have ANCA-GN + MTD.
We refer reviewer 1 to our response to the previous point and to the discussion in the article (page 8, line 224-229).
We analyzed CTD diagnosis among consecutive patients with biopsy proven AAV. In addition to the study cited by reviewer 1, other more recent studies have demonstrated a higher prevalence of ANCAs in both SLE and RA.
As an illustration, we refer reviewer 1 to the recent work of Tuner-Stokes et al done in a large cohort of consecutive patients with lupus nephritis (Kidney Int 2017, nov 92(5):1223-1231). In this work, authors found that 29 out of 203 patients, thus about 14% of the patients, had lupus nephritis with positives MPO or PR3 ANCAs.
We agree that frequency of ANCAs in RA patients is a quite rare condition. However, the very high prevalence of RA (0.5 to 1%, Scott DL et al, Lancet Sept 25;376(9746):1094-108) in contrast of AAV prevalence must be taken in consideration when analyzing frequencies.
Third, ANCA GN with CTD often correlates with disease severity and portends a poorer prognosis (Mittie K. Doyle. Current Rheumatology Reports 2006, 8:312–316), whilst Table 6 and the conclusion in this study suggest a comparable long-term prognosis between both groups. The analysis of renal biopsy did not show any significant difference between groups (Figure 3), which is inconsistent with previous studies. Considering the low event rate in a very small sample size study, thestatistical analysis is a critical issue.
We completely agree with Reviewer 1 that the development of vasculitis in patients with CDT is associated with a worse prognosis, but in reference to patients with CTD and without AAV. We do not question this point in our study.
In the present study, we didn’t analyze the impact of vasculitis on CTD. Indeed, we studied the impact of CDT on ANCA-GN outcome. Our study showed that CTD+ANCA-GN patients had a higher rate of thrombotic events and of minor AAV relapses, as compared to patients with AAV and without CTD. However, nor renal pathological involvement, nor hard outcomes (such as death and renal relapse) were different between groups. We acknowledge that this could be related to the size of the population, that may be underpowered to identify such a difference.
We also would like to underline that our study included more than 100 consecutive AAV patients and that it is the first study to report this kind of analysis.
Moreover, best to our knowledge, our work is the first to analyze and compare kidney pathological features between AAV patients with and without CTD.
In Table 2, the need for renal replacement therapy [n (%)]: 0 (0.0) in AAV + MTD group VS. 13 (15.5) in AAV group, the p value is non-significant (p=0.21)?I think AAV group is significantly associated with the need for renal replacement therapy. With respect to these cases with initial renal replacement therapy, renal relapses should not be comparable between groups.
We agree with reviewer 1 that this may be disappointing, however, using Exact Fisher test, which is the adequate test to use here, there is no statistical difference between groups.
We disagree with reviewer 1 comment suggesting that the need for RRT impacts relapse risk. Importantly, most AAV patients needing RRT initially recover renal function following immunosuppressive treatment initiation. Moreover, it has been shown that worse renal function (based on serum creatinine ou eGFR, and not RRT need) is associated with a decreased risk of AAV relapse in the literature. Unfortunately, the size of our population and the frequency of events do not allow to perform multivariate analyses in order to adjust on renal function and other parameters.
We added the following sentence and modified the original text to underline the limitation of our study in respect to non-renal relapses and size study limitations (page 9, line 248):
“The small size of the population and the low number of relapses did not allow to perform multivariate analysis and to adjust on gender and renal function. However, this observation suggests that maintenance regimens in patients with CTD should be carefully monitored and their intensity and duration eventually adjusted. As non-renal relapses involved mainly minor relapses, we also acknowledge that we could have misdiagnosed them as AAV relapses although they could reflect CTD activity.”
What are the definitions of renal and non-renal relapses? All patients with ANCA GN should receive strong immunosuppressive therapy and plasmapheresis/ plasma exchange, etc. Such standard treatments were specific to not only ANCA-GN but also CTD, as these agents were powerful and effective to prevent various relapses. Since renal relapses were comparable, please address why non-renal relapses were more frequent in ANCA GN + CTD group.
We acknowledge Reviewer 1 for this important comment.
We defined renal and non-renal relapses in Mat&Met section of the present manuscript (page 3, line 99). We agree that relapse need to be more accurately define in the manuscript.
Thus, we added the following precisions (page 3, line 101): “We individualized renal and non-renal relapses. Renal relapses were defined as relapses involving the kidneys, characterized by microscopic hematuria detection in association with a rise in serum creatinine of at least 25%. Non-renal relapses were defined as AAV activity in any other organ, solely or in combination, but excluding kidneys.”
We agree with reviewer 1 that strong immunosuppressive treatment is mandatory in systemic forms of AAV, notably those involving kidney. The use of plasma exchange is however not so systematic. Indeed, in our centers, until recently we used plasmapheresis in patients with severe kidney involvement (serum creatine >500 micromol/L at diagnosis as suggested by the MEPEX study and /or in cases with severe intra-alveolar hemorrhage or severe neurological involvement).
We don’t have definite explanations for the more frequent risk of non-renal relapses in CTD patients.
As discussed in the present manuscript, several explanations are discussed:
First: we wondered whether CTD patients had a less intensive immunosuppressive treatment. However, our observations didn’t support this possibility. Indeed, CTD patients were treated with similar attack treatments, as well as maintenance treatments. The use of plasma exchange was also similar between groups. Moreover, maintenance steroid treatment was also comparable. Thus, the higher risk of non-renal relapse is not likely due to differences in IS treatments. Second: different pathophysiological mechanisms in CTD patients as compared to patients without CTD. Indeed, one can suggest that some biological pathways are involved in patients with CTD, and account for a higher rate of non-renal AAV relapse Finally: we acknowledge that we cannot exclude that some patients had rather manifestations of CDT that we misdiagnosed as non-renal AAV relapses. This was underlined in the discussion of the present manuscript.
In light of Figure 2, the white bar was expressed as incidence ofAAV + CTD case number according to the year of diagnosis (biopsy-proven), and the black bar as AAV. However, 6 cases of AAV+CTD were diagnosed in 2016-2017. In these 6 cases, I wonder whether the follow up period is long enough to observe the events of deaths, cardiovascular outcomes, cancers and so on. Considering the low event rate in a very small sample size study, the statistical power is another issue.
We acknowledge reviewer 1 for this important comment.
We agree with reviewer 1 that the follow-up period is one of the limitations of our study. However, the follow-up period was not statistically different between groups (please see table 6, last line). The follow up period is shorter (even if not statistically different) in CTD patients and this clearly penalizes the CTD group. This may be one reason why we did not observe any other differences in prognosis between CTD and non CTD groups. However, despite a shorter follow-up, a higher risk of thrombotic events and a higher risk of non-renal relapses were still observed in CTD group.
We added the following sentence to the manuscript to underline these points (page 9, line 252): “The follow-up of patients, about 4.5 years, may also represent a limitation of the study.”
The consensus of current research indicates that dual presentation of ANCA-GN and CTD is rare, including authors' description in the Introduction section. However, the authors concluded that CTD is “frequently” associated with ANCA-GN. In my opinion, the term “frequently” should be deleted.
We agree with reviewer 1 that such an association is reported as rare in the literature, and that’s the reason why in the introduction of the present article, and according to the literature review, we stated that this is a rare association.
However, we report here that about 15% of consecutive patients with AAV have also a history of CTD. It is important to underline that our study is the first to report an estimation of such association, analyzing consecutive patients in a large population.
Thus, based on our data, we can not conclude that AAV + CTD is a rare event, but we agree with Reviewer 1 that we can also not conclude that it is a frequent condition given the prevalence of AAV.
According to reviewer comment, we modified the text as follows:
In the abstract, Page 1 line 32 : “Conclusion. Association between CTD and ANCA-GN is not a rare condition and predominantly affect females”
In the conclusion, page 9, line 275: “In conclusion, CTD in association with ANCA-GN diagnosis is not a rare condition, predominantly affecting women and without any significant differences in AAV presentation.”
Drug-induced ANCA-GN is an important issue. Is there any difference between groups about secondary causes related ANCA-GN?
We agree with reviewer 1 that drug induced ANCA-GN is an important issue. However, it is also a rare event in our experience and a challenging differential diagnosis. We didn’t observe any case of drug induced ANCA-GN in the patients of our registry who were included in the present study.
Reviewer 2 Report
The manuscript describes the incidence of connective tissue diseases (CTD) in a population of patients with ANCa-associated GN (ANCA-GN). The topic is quite intersting, given the possible overlapping into the pathogenic mechanisms of these disorders. The main issue of the present study is the relative low number of enrolled patients, especially when considering subgroups ( AR patients and so on), however the power of statistics needs confirmation.
Specific remarks:
Methods and table 2: the presence of organ involvement (cutaneous signs, heart etc) is listed in table 2. However the autours do not explain the definition of extra renal organs involvment. Please clarify.
Table 3: ANCA-GN patients with no CTD have a higher incidence of cryoglobulins and show no difference in FR compared with ANCA-GN with CTD patients. It is hard to interprete. It is possible that the low number of patients influences these results;
Table 6: long term outcomes. How did you consider long term outcomes? In the method section, we read a minimun period of 6 months of follow-up. Mean follow-up? 6 months seems a short time to consider long term outcomes. Please explain.
Author Response
Reviewer 2.
The manuscript describes the incidence of connective tissue diseases (CTD) in a population of patients with ANCa-associated GN (ANCA-GN). The topic is quite intersting, given the possible overlapping into the pathogenic mechanisms of these disorders. The main issue of the present study is the relative low number of enrolled patients, especially when considering subgroups ( AR patients and so on), however the power of statistics needs confirmation.
Specific remarks:
Methods and table 2: the presence of organ involvement (cutaneous signs, heart etc) is listed in table 2. However the autours do not explain the definition of extra renal organs involvement. Please clarify.
We acknowledge Reviewer 2 for this comment and agree that definition of organ involvement needs to be clarified in the manuscript. In fact, we used BVAS scale to define organ involvement, as usually done in manuscripts focusing on AAV. To clarify this point we added the following sentence to material and methods:
Page 2, line 90: “The activity of AAV and organ involvement at diagnosis and relapse were determined in reference to the Birmingham Vasculitis Activity Score (BVAS) 2003 [12]”
Table 3: ANCA-GN patients with no CTD have a higher incidence of cryoglobulins and show no difference in FR compared with ANCA-GN with CTD patients. It is hard to interprete. It is possible that the low number of patients influences these results;
We acknowledge reviewer 2 for this pertinent comment. We agree with reviewer 2 that this may be related to the size of the population. Moreover, rheumatoid factor was not screened in all patients as showed in table 3; and less frequently in non CTD patients as compared to the screening of cryoglobulin (the latter being frequently looked for in the context of glomerulonephritis).
However, we think the major explanation may be that a false positivity of RF is frequent in AAV. We refer reviewer 2 to the following and recent study showing a rate of 40% of false positivity: Moon JS et al, Clin rhumatol, 2018 Oct 37 (10) PMID:29119480.
Thus, we added the following sentence and the previous citation to the discussion to underline this important point:
Page 8-9, line 237-241: “As expected, significantly more patients were diagnosed with cryoglobulin positivity in CTD group, however no difference was observed according to rheumatoid factor positivity between groups. This may be explained by a high rate of false positivity of rheumatoid factor detected in AAV patients [30].”
Table 6: long term outcomes. How did you consider long term outcomes? In the method section, we read a minimun period of 6 months of follow-up. Mean follow-up? 6 months seems a short time to consider long term outcomes. Please explain.
We agree that the term “long term” may not be the most appropriate one. This point was also underlined by reviewer 1. Indeed, the follow-up period is about 4.5 years, not statically different between groups, which may be limited to analyse “hard outcomes” such as mortality.
The mean follow-up period (overall and for each group) is already given in table 6 (last line). We added the following sentence to the discussion section to underline this limitation of our study (page 9, line 252): “The follow-up of patients, about 4.5 years, may also represent a limitation of the study.”
Moreover, we deleted the term : “Long term” in the caption of table 6. We also deleted the term “long-term” in the manuscript.
Round 2
Reviewer 1 Report
Considering the following precisions (page 3, line 101): Renal relapses were defined as relapses involving the kidneys, characterized by "microscopic hematuria" detection in association with a rise in serum creatinine of at least 25%.
According to the literature review for definitions of renal relapses, renal relapses were commonly defined as new or worsening red blood cell casts and/or worsening proteinuria in association with renal function decline.
Author Response
We acknowledge reviewer 2 for this comment.
We modified the manuscript according to reviewer 2 suggestion.
Reviewer 2 Report
The revised Manuscript has addressed the majority of suggestions. The main limitation of the study , the relative low Number of enrolled patients, is still a Problem. I would state clearly in the discussion that the Sample size is a limitation of the study, and that the Statistic analysis among subgroups may be affected by this limitation.